# Association between Hip Center Position and Isokinetic Hip Muscle Performance after Anterolateral Muscle-Sparing Total Hip Arthroplasty

**DOI:** 10.3390/medicina58040538

**Published:** 2022-04-13

**Authors:** Hidetatsu Tanaka, Norikazu Yamada, Hiroaki Kurishima, Yu Mori, Toshimi Aizawa

**Affiliations:** 1Department of Orthopaedic Surgery, Japanse Redcross Sendai Hospital, 43-3, 2 Cho-me, Yagiyama Hon-cho, Taihaku-ku, Sendai 982-8501, Japan; yama@med.tohoku.ac.jp; 2Department of Orthopaedic Surgery, Tohoku University Graduate School of Medicine, 1-1 Seiryo-machi, Aoba-ku, Sendai 980-8574, Japan; marronile@gmail.com (H.K.); yu-mori@med.tohoku.ac.jp (Y.M.); toshi-7@med.tohoku.ac.jp (T.A.)

**Keywords:** total hip arthroplasty, hip center position, hip muscle strength

## Abstract

*Background and objectives:* The superior placement of the acetabular cup induced the delayed recovery of abductor muscle moment after total hip arthroplasty (THA) with a conventional posterior approach. The anterior-based muscle-sparing (ABMS) THA effectively reduces soft tissue damage, including muscles. The influence of hip center position on anterior-based muscle-sparing (ABMS) total hip arthroplasty (THA) for post-operative hip muscle strength was unclear. We evaluate whether the hip center position affects the recovery of hip muscle strength after ABMS THA. *Materials and Methods*: The study was performed as a retrospective cohort study, and included 38 hips in 38 patients that underwent primary ABMS THA. Muscle strength was measured using isokinetic dynamometry before the operation, and at 6 and 12 months after surgery. The horizontal and vertical centers of rotation (H-COR and V-COR), vertical shift (V-shift), leg length, and global femoral offset were determined radiographically in reference to a previous report. *Results:* A weak negative correlation was observed between abduction muscle strength at 6 months and V-shift; a V-shift more than 15 mm demonstrated significantly decreased abductor muscle strength at 6 months. *Conclusions:* The superior placement of the hip center caused delayed recovery of abductor muscle strength in hips with anterolateral minimally invasive THA. There seems to exist no biomechanical reason why the same should not also be the case for the muscle-sparing approach.

## 1. Introduction

Total hip arthroplasty (THA) provides relief from pain, restores functionality, and improves the quality of life for patients with hip disorders [1]. Recent studies reported THA with minimally invasive techniques is effective at reducing soft tissue damage, including muscles [2,3]. The anterior-based muscle-sparing (ABMS) THA is less commonly performed than the direct anterior approach (DAA) applied to the hip due to the clinical effectiveness, safety, very low dislocation rate, and excellent control of the leg length discrepancy of the latter [4,5]. Minimally invasive THA via the anterolateral approach in the supine position has yielded excellent clinical results [4].

The recovery of hip muscle strength after THA has been explored at many institutes, with considerable variability [6,7,8,9,10,11,12]. Fukushi et al. reported that the recovery of muscle strength after THA is especially affected by the implant position; superior placement of the hip center, more than 15 mm above the true hip center, delayed the recovery of abductor muscle moment after THA [13]. A reduction in global femoral offset after THA appears to have a negative association with the abductor muscle strength of the operated hip [14]. In most reports, conventional posterior or posterolateral approaches were used [8,9,13]. The aim of this study was, therefore, to evaluate whether the acetabular cup or hip position affects the recovery of hip muscle strength after THA using the anterior-based muscle-sparing approach.

## 2. Materials and Methods

### 2.1. Patients

This study was a single-center, retrospective cohort study, approved by our hospital’s institutional review board. The study considered for inclusion 55 hips from 55 patients that underwent unilateral primary THA with ABMS in the supine position at our institution between July 2017 and July 2018. The selection indication of THA with ABMS in this study was (1) no history of previous surgery on the affected hip, (2) primary osteoarthritis of the hip and osteonecrosis, and (3) secondary osteoarthritis of the hip with Crowe classification from 1 to 3 [5]. All hips were examined clinically and radiographically, and muscle strength was measured at 6 and 12 months after surgery. Of these, nine hips with severe OA of the unaffected side, six hips with THA of the unaffected side and, two hips with lower limb fracture, within the follow-up periods, were excluded from the analysis. Ultimately, the study included 38 hips in 38 patients. Table 1 shows the baseline demographic data, including age, sex, body mass index, diagnosis, and Crowe classification [15]. No post-operative complications, such as dislocation, infection, and/or implant loosening, have occurred in this series [16].

### 2.2. Assessments of Muscle Strength

Muscle strength was measured using a constant velocity muscle strength measuring machine, BIODEX SYSTEM 4 (Biodex Medical Systems, Shirley, NY, USA). The angular velocity was set to 60°/s, measured three times, and the maximum muscle contraction force value was adopted. The muscle strength evaluation used the peak torque value per body weight. Flexion strength was measured in the supine position; abduction muscle strength was measured in the lateral decubitus position. Flexion and abduction muscle strength were measured before the operation, and at 6 and 12 months after surgery. Biodex Medical Systems technology has acceptable mechanical reliability and validity [17].

### 2.3. Measurements of Radiographs

Conventional anteroposterior radiographs of the pelvis, including both hips and cross-table lateral views of the hip, were obtained to check the implant fixation. The pre-operative and post-operative leg length discrepancy (absolute value), and subsidence of the stem, were examined with anteroposterior radiographs. The global femoral offset (FO) was measured, with reference to a previous report by Sarwar et al. [14], by the addition of the distance between the longitudinal axis of the femur and the center of the femoral head, and the distance from the center of the femoral head to a perpendicular line passing through the medial edge of the ipsilateral teardrop point of the pelvis (Figure 1). The cup position was decided in reference to a previous report [18]. The center of the cup was determined by the intersection of two perpendicular diameters. The vertical center of rotation (V-COR) was defined as the vertical distance from the center of the femoral head to the inter-teardrop line (Figure 1). The horizontal center of rotation (H-COR) was defined as the horizontal distance to the teardrop, which is parallel to the inter-teardrop line (Figure 1) [18]. The vertical shift (V-shift) was defined as the difference in V-COR between the affected hip and the contralateral normal hip [18].

### 2.4. Statistical Analyses

Statistical analyses were performed using the EZR (Saitama Medical Centre, Jichi Medical University, Saitama, Japan), which is a graphical user interface for R (The R Foundation for Statistical Computing, Vienna, Austria) [19]. Specifically, it is a modified version of the R commander designed to add statistical functions, and is frequently used in biostatistics. Wilcoxon signed-rank tests were used to compare the pre-operative and post-operative muscle strength and radiographic data. The comparisons of continuous variables were compared using the Mann–Whitney U test. Spearman’s rank correlation coefficient was used to assess the correlations between muscle strength and hip center position. Statistical significance was defined as *p*-values < 0.05.

## 3. Results

Table 2 shows the hip flexion and abductor muscle strength at each time point. Both muscle strengths were significantly improved after the operation, and the acquired muscle strength was maintained at 6 and 12 months (Table 2).

Table 3 indicates pre- and post-operative leg length discrepancy and global FO. Pre-operative leg length was significantly different between Crowe classifications; however, post-operative leg length was restored relative to the pre-operative condition. There were no significant differences in global FO. As for the post-operative cup position, V-COR and V-shift were significantly higher in Crowe 2 (Table 3). H-COR had no significant differences.

Table 4 shows the comparison of the hip flexion and abductor muscle strength between Crowe classification. Pre-operative abductor muscle strength of Crowe 1 was significantly lower than that of Crowe 2. There were no significant differences between pre-operative flexion muscle strength and post-operative muscle strength. A weak negative correlation was observed between abduction muscle strength at 6 months and V-shift (Figure 2 and Figure 3). Regarding the small group sub-analysis of V-shift, a V-shift of more than 15 mm demonstrated significantly decreased abductor muscle strength at 6 months (*p* = 0.006, Dunnett’s test) (Figure 4). However, no significant differences were observed among the four subgroups in terms of V-shift at 12 months post-operation (*p* = 0.272) (Figure 4).

## 4. Discussion

The results of this study show an improvement in muscle strength after THA, given that the hip flexion and abduction muscle strength increased by about 1.5 to 1.7 times at 6 months after THA, and the acquired muscle strength was maintained at 12 months. These findings show that peak hip muscle recovery occurred less than 6 months after ABMS THA. Previous studies reported that hip muscle weakness recovered within 6 to 24 months after THA [6,7,8]. For example, abduction muscle strength decreased by about 26% from the pre-operative baseline 1 month after surgery, and recovered to the same pre-operative level 3 months after surgery [8]. Rasch et al. demonstrated a slow but full recovery of muscles acting around the knee, but there was still a deficit in hip muscle strength 2 years after THA [6]. 

Several radiographic factors, such as hip center position and global FO, affected the improvement in abductor muscle strength after THA [13,14]. Our data show that V-shift most strongly affected the improvement in abductor muscle strength. These findings are consistent with a previous study [13]. Delayed recovery of abductor muscle strength was observed in hips with the superior placement of the hip center, although anterolateral minimally invasive THA is effective in reducing soft tissue damage including hip muscles. Therefore, surgeons should try to avoid placing the acetabular cup superior position. On the other hand, in cases with severe DDH, superior placement of a cementless acetabular cup is often accepted to have insufficient acetabular coverage [20]. The global FO in our study had a weak negative correlation with improvement in abductor muscle strength after THA. 

Our data included cases with developmental dysplasia of the hip (DDH), and 13 of 38 cases were Crowe classification type 2. The pre-operative abductor muscles with Crowe type 2 were significantly lower compared to Crowe type 1. Although the hip center was significantly higher in hips with Crowe type 2, the post-operative abductor muscles exhibited no significant differences between the hips in Crowe type 1 and 2. In addition to the hip center, various factors, such as global FO, and anteversion or antetorsion of the femur, are intricately related, and further study is needed to resolve the factors concerning the recovery of, and improvement in, muscle strength after THA.

Our study has several limitations. The first limitation is that our study was focused on the gluteus muscle load arm, but did not consider the force arm. The abduction moment that the hip abductors must develop is determined by the relationship between the load arm of the body weight and the force arm of the gluteal muscles; for the load arm, the acetabular offset can be used, but for the force arm of the gluteal muscles, only the distance of the gluteal muscles from the hip center of rotation can be used. Further study focusing on the force arm is desirable. The second is that the radiographic parameter was evaluated using plain radiography. Further study using computed tomography is required to evaluate hip position and femoral anteversion on the transverse and sagittal plane. The third limitation is that this study was a retrospective, single-center study and, as such, it had no control group. The number of cases with severe DDH, such as Crowe type 2 or 3, were fewer. The fourth limitation is the inadequate evaluation of the spine and legs, except for around the hip joint, which is important because hip flexion and abduction involve movement of the trunk. Finally, the reproducibility of the radiographic measurements was not evaluated by assessing the degree of agreement between the measurements of multiple evaluators.

## 5. Conclusions

This study investigated whether the position of the acetabular cup affects the recovery of hip muscle strength after THA using the anterior-based muscle-sparing approach. The superior placement of the hip center caused the delayed recovery of abductor muscle strength in hips after THA with ABMS. Surgeons should be aware of the precise placement of the acetabular cup in THA with ABMS.

## Figures and Tables

**Figure 1 medicina-58-00538-f001:**
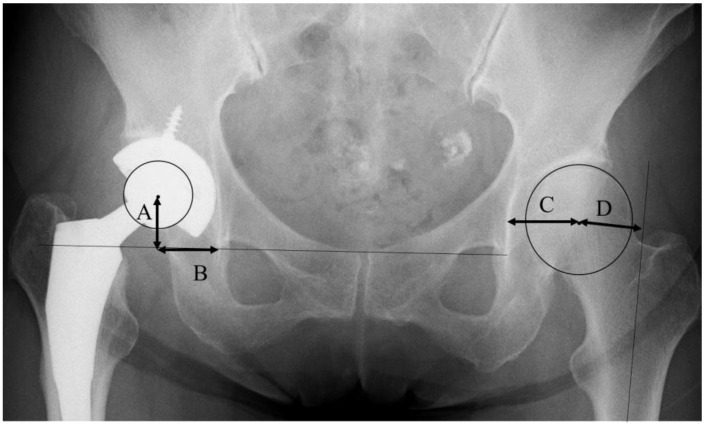
The definition of radiographic parameters with reference to a previous report [13,14,17]. (A) Vertical center of rotation (V-COR). (B) Horizontal center of rotation (H-COR). (C + D) Global femoral offset (FO).

**Figure 2 medicina-58-00538-f002:**
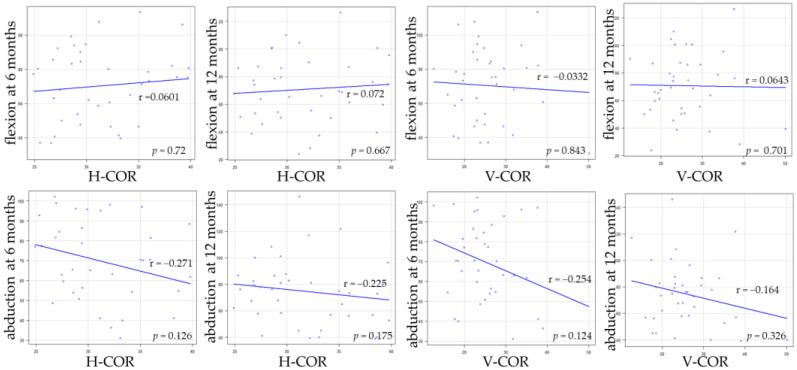
The correlations between the hip muscle strength and H-COR, V-COR.

**Figure 3 medicina-58-00538-f003:**
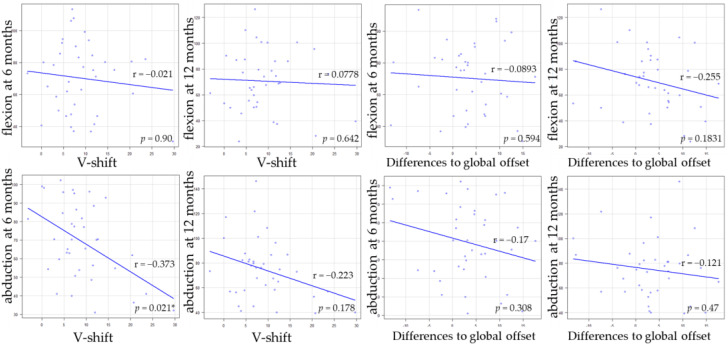
The correlations between the hip muscle strength and V-shift, differences in global offset. * *p* < 0.05.

**Figure 4 medicina-58-00538-f004:**
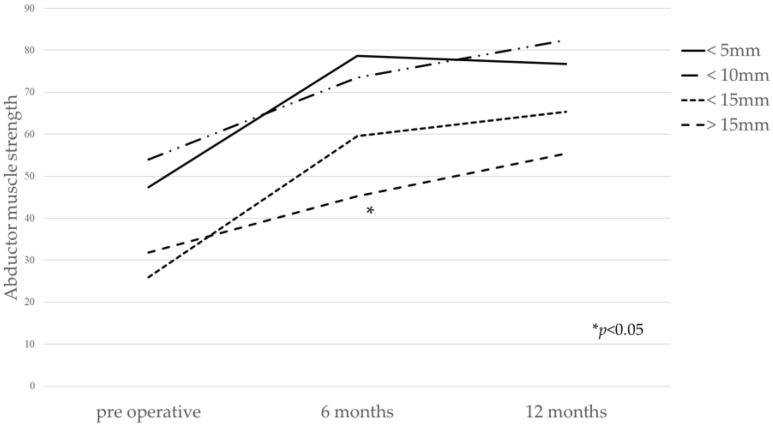
The relation between V-shift and abductor muscle strength at each point. V-shift of more than 15 mm demonstrated significantly decreased abductor muscle strength at 6 months (*p* = 0.006, ANOVA with post hoc analysis using Dunnett’s test).

**Table 1 medicina-58-00538-t001:** Patient demographic data.

Number of patients	38
Age at time of operation, mean ± SD (range)	66.5 ± 7.9 (52–87)
Gender Female:Male, no. of patients (%)	32 (84.2):6 (15.8)
Body mass index	24.9 ± 3.5
Diagnosis	
Osteoarthritis	38
Crowe classification	
1	25
2	13
3	0
4	0

Data represent mean ± standard deviation.

**Table 2 medicina-58-00538-t002:** Muscle strength pre-operation, and at 6 and 12 months post-operation.

	Pre-Operation	At 6 Months	*p*-Value	At 12 Months	*p*-Value (for Pre-Operation)	*p*-Value (for Six Months)
flexion	42.3 ± 22.0	70.3 ± 21.6	0.000 *	70.7 ± 23.4	0.000 *	0.556
abduction	44.0 ± 27.0	69.3 ± 20.4	0.000 *	74.8 ± 23.9	0.000 *	0.0651

Data represent mean ± standard deviation. * *p* < 0.05. Wilcoxon signed-rank test was performed to compare scores.

**Table 3 medicina-58-00538-t003:** Post-operative hip position and leg length change by Crowe classification.

	Crowe Classification	Value	*p*-Value
pre-operative leg length discrepancy (mm)	1	−5.6 ± 4.9 (−19.4~−3.3)	
	2	−17.1 ± 9.8 (−31.3~−4.1)	0.000 *
post-operative leg length discrepancy (mm)	1	0.7 ± 3.8 (−10.5~7.6)	
	2	0.4 ± 9.1 (−14.3~8.8)	0.925
pre-operative global FO (mm)	1	77.8 ± 7.2 (63.0~92.4)	
	2	74.8 ± 5.6 (62.3 ~83.5)	0.275
post-operative global FO (mm)	1	79.4 ± 7.8 (64.3~102.1)	
	2	77.9 ± 7.6 (65.8~90.4)	0.236
V-COR (mm)	1	23.0 ± 4.4 (12.6~31.9)	
	2	31.7 ± 8.2 (18.5 ~45.1)	0.000 *
H-COR (mm)	1	29.9 ± 4.2 (24.9~39.7)	
	2	33.1 ± 4.7 (25.3~39.8)	0.484
V-shift (mm)	1	6.2 ± 4.3 (−3.1~16.5)	
	2	12.0 ± 7.0 (6.7~22.7)	0.000 *

Data represent mean ± standard deviation (min/max). * *p* < 0.05. Mann–Whitney U test was performed to compare values between Crowe classification.

**Table 4 medicina-58-00538-t004:** Muscle strength between the Crowe classification at each period.

	Crowe Classification	Pre-Operation	*p*-Value	At 6 Months	*p*-Value	At 12 Months	*p*-Value
flexion	1	42.2 ± 21.0		73.3 ± 18.9		67.6 ± 20.6	
	2	36.8 ± 21.0	0.498	80.3 ± 26.1	0.841	76.2 ± 27.8	0.633
abduction	1	57.4 ± 23.9		70.5 ± 17.2		77.8 ± 22.4	
	2	32.8 ± 22.8	0.0157 *	61.9 ± 23.4	0.113	72.8 ± 25.8	0.295

Data represent mean ± standard deviation. * *p* < 0.05. Mann–Whitney U test was performed to compare values between Crowe classification.

## Data Availability

The data that support the findings of this study are available on request from the corresponding author.

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
