# Peer review of "Association between Hip Center Position and Isokinetic Hip Muscle Performance after Anterolateral Muscle-Sparing Total Hip Arthroplasty"

_medicina, 2022, doi:10.3390/medicina58040538_

Round 1

Reviewer 1 Report

Dear Authors,

please find my comments in the reviewed PDF-file of your manuscript.

Author Response

Point-by-point responses to reviewers' comments

  We greatly appreciate the reviewers’ careful and kind reviews of our manuscript (medicina-1648747), entitled “Association between hip center position and isokinetic hip muscle performance after anterolateral muscle sparing total hip arthroplasty”. Here are our responses to the reviewers’ comments.

In addition, the research design, description of methods, presentation of results and conclusion need to be improved.

Response.  We greatly appreciate the reviewer's careful and kind comments.

  1. Conclusion is common knowledge from traditional standard approaches; There seems to exist no biomechanical reason why the same shouldn't be the case for the ALMI approach too

Response. Thank you for your comment. We added the sentence described below.

Line 23-26: Conclusions: The superior placement of the hip center caused delayed recovery of abductor muscle strength in hips with anterolateral minimally invasive THA. There seems to exist no biomechanical reason why the same shouldn't be the case for the muscle-sparing approach too.

  1. suggestion: with respect to your final inclusion criteria you might prefer to use the term "selection criteria" at this point because you are listing criterias that have to be fulfilled to be selected for this study rather than indications for total hip arthroplasty;

Response. We agreed with the reviewer's comment. Thank you for your comment. We changed the sentence described below.

Line 52: The selection indication of THA~

  1. please inform the reader which muscles have been tested and how you did position the patient in particular for abduction muscle strenght measurement

Response. Thank you for your comment. We added the measured position described below.

Line 68-70: The flexion strength was measured in the supine position; abduction muscle strength was measured in the lateral decubitus position.

  1. what does this mean: "leg lenght was improved"???was it restored? equalized with the opposite side? or ?

Response. We have changed the phrases in Table 3.

Line 110-113: Table 3 indicates pre- and post-operative leg length discrepancy and global FO. Pre-operative leg length was significantly different between Crowe classification; however, post-operative leg length was restored compared to pre-operative condition.

  1. Table 5 contains only a set of statistical data that are hard to interprete;

I recommend including these data in Fig. 2 to make the diagram more informative and omitting Table 5

Response. Thank you for your comment. I changed the contents of Table 5 to Figure 2. I believe that it's easy to understand visually for readers.

Line 119-124: A weak negative correlation was observed between abduction muscle strength at 6 months and V-shift (Fig. 3, 4.). With the small group sub-analysis of V-shift, V-shift of more than 15 mm demonstrated significantly decreased abductor muscle strength at 6 months (p = 0.006, Dunnett’s test) (Fig.5). However, no significant differences were observed among the 4 subgroups of V- shift at 12 months post-operatively (p = 0.272) (Fig.5).

Figure 3. The correlations between the hip muscle strength and H-COR, V-COR.

Figure 3. The correlations between the hip muscle strength and V-shift, differences of global offset.

  1. what point???

this diagram shows the time-dependent course of the abductor muscle strength, the curve parameter is the vertical shift; please describe in a more informative way what kind of data you are presenting

Response. We appreciate the reviewer's comment. I'm sorry to confuse you.

I added an asterisk in Figure 5, and interpretation. Please check again.

Line 128-130: Figure 5. The relation between V-shift and abductor muscle strength at each point. V-shift of more than 15 mm demonstrated significantly decreased abductor muscle strength at 6 months (p = 0.006, ANOVA post hoc by Dunnett’s test).

  1. suggestion:

put these sentences at the beginning of the discussion paragraph because this is the most important message of your study and continue by saying "whereas previous studies reported ...

Response. We agreed with the reviewer`s suggestion. We corrected.

  1. it is unclear what these two sentences are supposed to express in the discussion

Response. Thank you for your pointing out. In our study, fixation of cup was not evaluated (there was no loosening case). We delete these sentences because our study the relationship between muscle strength measurement and installation height.

  1. From my point of view, there is a major methodological weakness in their study: the abduction moment that the hip abductors must develop is determined by the relationship between the load arm of the body weight and the force arm of the gluteal muscles; for the load arm, the acetabular offset can be used, as you did; but for the force arm of the gluteal muscles, only the distance of the gluteal muscles from the hip center of rotation can be used;

You should keep in mind that the torque that the gluteal muscles are developing is decisive for the ability to walk, not the force per se that the gluteal muscles develop for abduction! This force is always the same at time x postoperatively and fortunately this force is increasing along the time-line;

The decrease in gluteal muscle force observed with increasing V-shift is caused by the shortening of muscle fibers, i.e. what is called "active insufficiency" of a muscle; the correlation with this kind of geometric modification is a co-correlation;

    You should indeed revise the whole concept of your study from the biomechanical aspect!

Response We greatly appreciate the reviewer's careful and kind reviews.

We interpreted that our study is focusing on the gluteus maximus load arm but not considering the force arm. Focusing solely on abduction strength is not enough. I should keep in mind that the torque that the gluteal muscles are developing is decisive for the ability to walk, not the force power that the gluteal muscles develop for abduction. However, it will be difficult to evaluate the force arm evaluation from now on and to get the data. So, I added your point of view to the limitation. This research focuses on the load arm of the gluteus maximus, and the force arm is a topic for me in the future. We added the sentence to limitation described below.

Line 167-173: The first limitation is that our study was focusing on the gluteus muscles load arm but not considering the force arm. The abduction moment that the hip abductors must develop is determined by the relationship between the load arm of the body weight and the force arm of the gluteal muscles; for the load arm, the acetabular offset can be used, but for the force arm of the gluteal muscles, only the distance of the gluteal muscles from the hip center of rotation can be used. Further study is desirable focusing on force arm.

  1. missing space

Response. We agreed with the reviewer`s suggestion. We corrected.

Reviewer 2 Report

Dear authors the topic is very interesting.

Your methods are well described and the discussion is supported by your results.

As regard the results and discussion, it will be interesting for reader if you link these measurements to complications. Did you observed luxation or mobilization of your implants? What is the satisfaction grade of the patients in relation to muscle strenght recovery?   

Furthemore you may cite the following article in which Authors described a temible complication due to aseptic loosening of acetabular cup. The malposition of acetabular cup causes inflammatory reaction, metallosis and pseudotumor

-First case report of vanadium metallosis after ceramic-on-ceramic total hip arthroplasty

Journal of Biological Regulators and Homeostatic AgentsVolume 27, Issue 4, Pages 1063 - 1068October 2013

Pesce V. et al

Author Response

Point-by-point responses to reviewers' comments

  We greatly appreciate the reviewers’ careful and kind reviews of our manuscript (medicina-1648747), entitled “Association between hip center position and isokinetic hip muscle performance after anterolateral muscle sparing total hip arthroplasty”. Here are our responses to the reviewers’ comments.

Reviewer 2: Your methods are well described and the discussion is supported by your results.

Response.  We greatly appreciate the reviewer's careful and kind comments.

  1. As regard the results and discussion, it will be interesting for reader if you link these measurements to complications. Did you observed luxation or mobilization of your implants? What is the satisfaction grade of the patients in relation to muscle strenght recovery?

Response. Thank you for your comment. We added the post operative complication described below. This study focuses on between muscle strength recovery and the center of the hip. The clinical score was satisfactory, however no statistical examination have been conducted on recovery of muscle strength and improvement of clinical score in this series.

Line61-62: No postoperative complications such as dislocation, infection, and implant loosening have occurred in this series [16].

  1. Furthemore you may cite the following article in which Authors described a temible complication due to aseptic loosening of acetabular cup. The malposition of acetabular cup causes inflammatory reaction, metallosis and pseudotumor

First case report of vanadium metallosis after ceramic-on-ceramic total hip arthroplasty

Journal of Biological Regulators and Homeostatic AgentsVolume 27, Issue 4, Pages 1063 - 1068October 2013, Pesce V. et al

Response. Thank you for your comment. We added your paper.

Line61-62: No postoperative complications such as dislocation, infection, and implant loosening have occurred in this series [16].

Line242-244: 16. Pesce V, Maccagnano G, Vicenti G, Notarnicola A, Lovreglio P, Soleo L, Pantalone A, Salini V, Moretti B. First case report of vanadium metallosis after ceramic-on-ceramic total hip arthroplasty. J Biol Regul Homeost Agents 27(4): 1063, 2013
